

# Characteristics of dew on leaves of typical plants in Loess Hill and Gully Region of China

Zhifeng Jia[1, 2, 3], Ge Li[1, 2, 3], Cheng Jin[1, 2, 3], Jun Xing[4], Yingjie Chang[1, 2, 3], Pengcheng Liu[5], Danzi Chen[6]

[1]School of Water and Environment, Chang'an University, Xi'an 710054, China
[2]Key Laboratory of Subsurface Hydrology and Ecological Effects in Arid Region, Ministry of Education, Chang 'an University, Xi'an 710054, China
[3]Key Laboratory of Eco-hydrology and Water Security in Arid and Semi-arid Regions of the Ministry of Water Resources, Chang'an University, Xi'an 710054, China
[4]School of Civil Engineering, Chang'an University, Xi'an 710054, China
[5]Xi'an Water (Group) Lijiahe Reservoir Management Company, Xi'an 710055, China
[6]No.203 Research Institute of Nuclear Industry, China Nuclear Geology, China National Nuclear Corporation, Xi'an 710100, China

*Correspondence to*: Zhifeng Jia (409538088@chd.edu.cn)

**Abstract.** Dew plays an important role as a non-precipitation input to the hydrological cycle and vegetation recovery in semi-arid ecosystems. To clarify the characteristics of dew on plant leaves, typical plants such as Tribulus, Hippophae and Elm were selected in the hilly and gully region of the Loess Plateau in China. The dew amount of typical plant leaves by manual weighing measurement was combined with automatic observation data from leaf wetness sensors (LWS) further to realize the automatic observation of dew on the leaf surface. The results showed that the cumulative dew amount on Tribulus, Hippophae and Elm from May to October in 2022 was in the order of Tribulus (17.46 mm) > Hippophae (11.14 mm) > Elm (5.88 mm), and the difference in the amount was mainly related to the leaf inclination angle and the microstructure of the leaf blade. Dew mainly occurred from 22:00 to 9:00 the next day on a daily scale, and was mainly concentrated in July–October during the year. The dew amount on the leaves of the three plants was less than the amount of rainfall, but the dew frequency was higher than that of rainfall. It was more favorable for the dew formation when the relative humidity is greater than 80.0 %, the difference between air temperature and dewpoint was less than 2 ℃, and the wind speed was less than 1.0 m/s. In addition, there was more dew on the first day after a rainfall than on the day before due to sufficient moisture.

## 1 Introduction

In arid and semi-arid zones, any source of moisture other than rainfall may have a positive impact on ecosystems (Uclés et al., 2013; Kidron and Temina, 2017; Kaseke and Wang, 2018). As a non-precipitation water resource, dew can provide water recharge to the ecosystem due to its frequent and stable occurrence (Zhang et al., 2015), and plays a pivotal role in the balance of the water cycle in arid and semi-arid areas.

China's arid and semi-arid regions cover an area of more than $2.56 \times 10^6$ km$^2$, with fragile ecological environments and a shortage of water resources that is a major constraint on the growth of crops and natural vegetation (Kidron, 2000; Kidron and



Starinsky, 2019; Tsafaras et al., 2022; Jia et al., 2023). Dew is the product of water vapor condensation in the near-surface air, formed by radiation heat dissipation of water vapor on the surface of vegetation or soil after the temperature drops to the dew

point during the night with little or no wind (Agam and Berliner, 2006; Wang et al., 2017; Zhang et al., 2019; Feng et al., 2021). According to the different condensation surfaces, dew can be divided into plant leaf dew and soil dew, the former source of water vapor is mainly water vapor in the air and water vapor exchanged by plant respiration, and the latter mainly consists of water vapor in the air and water vapor in shallow soil. In this study, plant leaf dew was analyzed. Multiple studies have shown that plant leaves can absorb dew (Stone, 1957; Liu et al., 2020), and transport it to the plant root system for utilization

(Eller et al., 2013); The duration of dew on the canopy of Artemisia oleifera was longer than that on the canopy of Salix salicifolia in the Mao Wusu Desert, and that the differences in the duration of canopy and sub-canopy dew were dependent on plant species (Guo et al., 2022); Interestingly, the microstructure of plant leaves can also influence dew collection and utilization. For example, uniformly distributed conical spines on cactus stems are able to collect dew in a directional manner (Ju et al., 2012). The presence of hairy tips on the leaf blade contributes to the utilization of dew and precipitation by the

toothed-ribbed ruderal crust and enhances the adaptability of the moss crust to arid environments (Tao and Zhang, 2012).
At present, the observation method of dew is still being explored. It is mainly divided into the observation of dew duration, dew quantity and dew source (Agam and Berliner, 2006). Dew duration is mainly measured by leaf wetness sensor method (Scherm and van Bruggen, 1993), impedance grid method (Pedro and Gillespie, 1981), etc.; Dew amount is mainly measured by Duvdevani method (Subramaniam, 1983), Hiltner dew balance meter method (Zangvil, 1996), microosmography method

(Xiao et al., 2009), etc.; the source of dew is mainly observed by isotope technology tracer method (Burgess and Dawson, 2004; Kim and Lee, 2011; Goldsmith et al., 2017; Berry et al., 2019). Among them, the leaf wetness sensor (LWS) is more accurate in the observation of dew duration and the dew amount can be obtained through calibration ( Jia et al., 2019; Binks et al., 2019; Gao et al., 2020; Gao and Wang, 2022; Khabbazan et al., 2022). In this paper, typical site was selected in the loess hilly and gully region of China, and manual dew observation to three typical plants was used to calibrate the dew amount

measured by LWS. The characteristics of leaf dew in three typical plants were analyzed, so as to clarify the characteristics of water vapor condensation on plant leaf in the Loess Plateau of China and promote the understanding of the hydrological cycle in the arid region.

## 2 Materials and methods

### 2.1 Experimental site

The experimental station is located in Ansai Experimental Station, Institute of Soil and Water Conservation, Ministry of Water Resources, Chinese Academy of Sciences, Ansai District, Yan'an City, Shaanxi Province, China. The study area is a typical loess hill and gully area with a mid-temperate continental semi-arid monsoon climate. (Fig. 1). The area has an altitude of 1068–1309 m, average annual temperature of the region is 8.8 °C, the average annual precipitation is 500 mm (Jia et al., 2023a), the average annual evapotranspiration is 1000 mm, the frost-free period is 157 d (Zhao et al., 2017), the annual sunshine hours





are 2352–2573 h; Common plants at the Experimental Station include Elm, Periploca sepium, Artemisia gmelinii, Heteropappus altaicus, Hippophae, Artemisia scoparia, Robinia psendoacacia, and Tribulus (Fang et al., 2016).

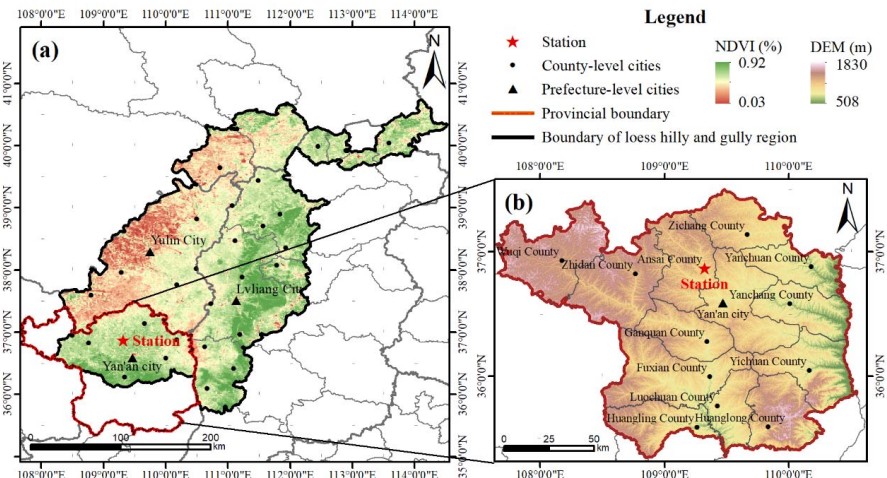

**Figure 1. Location of the experimental site (Jia et al., 2023b): (a) Geographic location of the loess hill and gully area, (b) Geographic**
**location of the study area.**

### 2.2 Typical plants selection

Tribulus, Hippophae and Elm near the experimental site were selected as typical plant representatives. Tribulus with a heigh of about 0.1 m tall is an annual growing herb, even-numbered pinnately compound leaves, leaflets opposite, and leaf surface densely covered with long tomentum. The growth period of Tribulus can be divided into seedling, stem and leaf growth, bud

pregnancy, flowering, fruit ripening and wilting period, usually from mid-May to the end of October; The growth height of Hippophae is about 1.5 m. Its leaves are pubescent, narrow, and lanceolate, with a silver-gray color (Ma et al., 2023). The growth period of Hippophae can be divided into germination stage, leaf spreading stage, shoot growth stage, flowering stage, fruit maturity stage. Germination stage is in mid-April, and the leaves gradually change from green to gray-black from September to November; The elm tree has a height of about 25 m. It is a deciduous tree with elliptic-ovate leaf blades, smooth

and glabrous on the surface, pubescent on the underside of the leaf when young and then glabrous, and biserrate or uniserrate on the margins. The growth period of the elm tree is divided into four stages: germination stage, growth stage, maturity stage and declining stage budding. It usually begins in April and ends around October each year. In summary, based on the growth period of the three plant leaves, the time for analyzing the dew of typical plant leaves in this experiment was from May to October.



## 2.3 Experimental design

The distribution of typical plants such as Tribulus, Hippophae and Elm leaves near the test site is shown in Fig. 2a. The experiment was laid out next to the three plants, and three leaf wetness sensors (LWS) with 0.02 mm resolution were respectively installed at 0.2 m, 0.6 m and 1.5 m above the ground as possible as the height of the three plants. Atmospheric temperature and relative humidity (RH) were measured by ATMOS14 sensor with 0.1 ℃ and 0.1 % RH resolution, placed at 0.2 m above the ground. Wind speed and direction were measured by the WSD01 anemometer with 1 mph and 1° resolutions, placed at 1 m above the ground. Precipitation was measured by the ECRN-100 rain gauge installed at 1.6 m at a horizontal distance of 32 m from the site, with a resolution of 0.2 mm. All sensors were automatically recorded by an EM50 data collector at 30-min intervals. All the instruments above were manufactured by Decagon Devices (Pullman, WA, USA). The installation schematic is shown in Fig. 2b, and the real view is shown in Fig. 2c. The observation period was from September 1, 2021, to October 31, 2022.

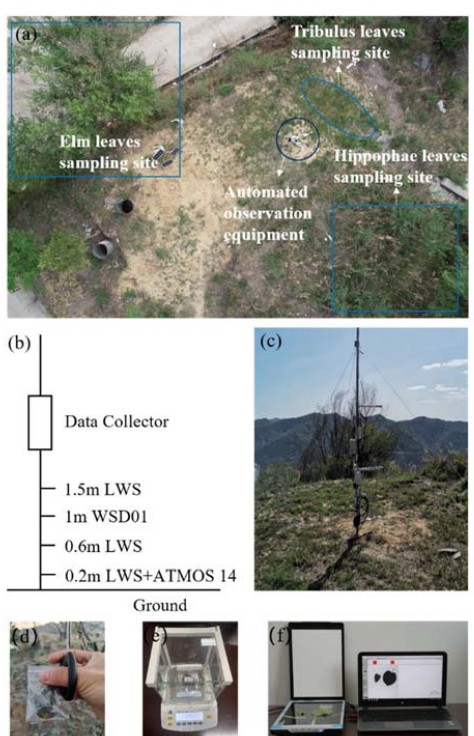

**Figure 2. Experimental layout and process. (a) Plant distribution map; (b) Design diagram of equipment installation; (c) Actual view of the equipment; (d) Leaf sampling; (e) Leaf weighing; (f) Determination of leaf area.**



Manual field experiments in September-October 2021 and September-October 2022 were conducted concurrently with the automatic observations to calibrate the LWS. Leaves from three plants were picked at 6:30, 7:00, 7:30, 8:00 in the morning and brought back to the lab for weighing. To minimize the effect of height on dew, the sampling heights of Tribulus, Hippophae and elm were 0.02 m, 0.6 m and 1.5 m, respectively. The number of leaves was 5 for Tribulus and Elm, and 10 for Hippophae. The mass of the ziplock bag was weighed and recorded as $m_2$ prior to sampling, and then filled into the ziplock bag immediately

after sampling. The sealed bag was placed under the blade using tweezers and scissors to avoid the loss of dew during sampling. After sampling was completed, the sealed bag containing the leaves was returned to the laboratory and weighed using a high-precision balance with 0.001g resolution and recorded as $m_1$. The leaves were then removed, and the dew on the leaves was blotted up using absorbent paper. The plant leaves without dew were weighed and recorded as $m_3$. Plant leaf area was measured using SYS-Leaf1000 leaf image analyzer manufactured by Epson (China) Co. with 0.01 cm$^2$ resolution. The measure procedure

is shown in Fig. 2d-f. The dew amount per unit area can be calculated by Eq. (1). The number of manual observation days was 61 days during the test.

$$W_d = \frac{1000\left(m_1 - m_2 - m_3\right)}{\sum\limits_{i=1}^{n} S_i}\rho,$$  (1)

Where $W_d$ is the leaf dew amount per unit area (mm), $m_1$ is the weight of the sealed bag containing the leaf dew (g), $m_2$ is the weight of the sealed bag (g), $m_3$ is the dry weight of the leaf (g), $\rho$ is the density of water (1 g/cm$^3$), $S_i$ is the area of the $i$th

plant leaf, and $n$ is the number of leaves, where Tribulus and Elm $n$=5 and Hippophae $n$=10.

### 2.3.1 Calibration of leaf wetness sensor

The leaf wetness sensors (LWS) were used to measure the dew, which has a surface with radiative and thermodynamic properties similar to those of real leaves, and a hydrophobic surface coating, similar to the hydrophobic cuticle of real leaves. A relationship between sensor micro voltage ($x$) and the thickness of the water layer ($W_d$) on the sensor surface was obtained

by Jia et al (2019), as shown in Eq. (2)

$$W_d = ax^b,$$  (2)

Where $W_d$ is the dew amount per unit area (mm), $x$ is the voltage value output from the leaf wetness sensor (mv), $a$ and $b$ are the fitting parameters.

Tribulus leaves, Hippophae leaves and Elm leaves were respectively corresponded to the LWS at 0.2 m, 0.6 m, and 1.5 m

according to their growth heights. Calibration curves for three typical plants were obtained by analyzing the manually observed dew and LWS data as shown in Fig. 3. Corresponding to the parameters, respectively, Tribulus $a = 4.1 \times 10^{-7}$, $b = 1.913$, $R^2 = 0.438$; Hippophae $a = 3.5 \times 10^{-10}$, $b = 2.883$, $R^2 = 0.475$; Elm: $a = 1.5 \times 10^{-11}$, $b = 3.316$, $R^2 = 0.426$.





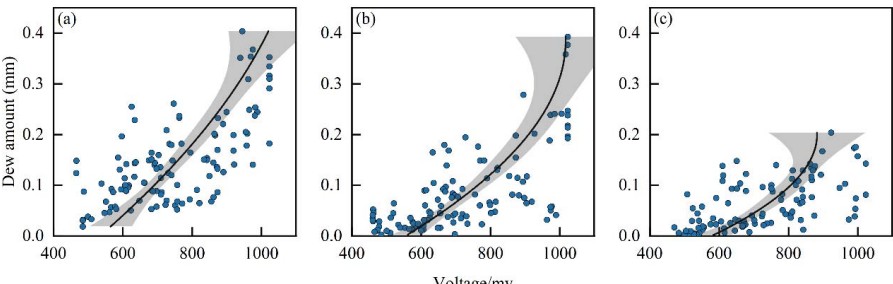

**Figure 3. Plant leaves dew with LWS value: (a) Tribulus leaves, (b) Hippophae leaves, (c) Elm leaves.**

### 2.3.2 Dewpoint temperature

Dewpoint temperature is calculated using the Lawrence equation (Lawrence, 2005).

$$T_d = \frac{B_1\left[\ln\left(\dfrac{H_R}{100}\right) + \dfrac{A_1 T_a}{B_1 T_a}\right]}{A_1 - \ln\left(\dfrac{H_R}{100}\right) - \dfrac{A_1 T_a}{B_1 T_a}},$$ (3)

Where $T_a$ and $T_d$ are air temperature and dewpoint temperature (℃), $H_R$ is relative humidity of the air (%), $A_1$ and $B_1$ are the coefficients recommended by Alduchov and Eskridge: $A_1$=17.625, $B_1$=243.04℃ (Alduchov and Eskridge, 1996).

### 2.3.3 Relevant definitions

(1) Daily dew

Since the dew occurrence is affected by the fluctuation of meteorological factors, dew occurs only during part of the day. Therefore, the daily dew amount is defined as the sum of the dew amount in the period when dew occurs, i.e.

$$D_d = \sum_{i=1}^{n}\left(W_{d\max} - W_{d\min}\right),$$ (4)

Where $D_d$ is the daily dew amount (mm), $i$ is the order of periods, $n$ is the total number of periods per day (n = 24 if the period length is 1 hour), $W_{dmax}$ is the maximum dew amount monitored in the $i$th period (mm), $W_{dmin}$ is the minimum dew amount monitored in the $i$th period (mm). When the observed value in the period changes from small to large, it is regarded as the process of dew cumulative increase, and when the observed value in the period changes from large to small, it is regarded as the process of dew evaporation.





(2) The time for calculating the dew amount is from 16:00 on the day to 16:00 on the next day. At the same time, the rainfall period data was excluded, that is, the dew amount during the rainfall is 0.

(3) The frequency of dew occurrence is the ratio of the number of days with dew to the total number of days.

(4) The dew duration is the length of time dew appears on the leaf surface.

### 2.3.4 Data processing

One-way analysis of variance (ANOVA) was used to analyze the daily dew amount in the leaves of three plants, which was completed by SPSS 25.0 software and illustrated by Origin2023.

## 3 Results

### 3.1 Daily variations of dew on plant leaves

Three typical days on August 3, September 24, and October 11, 2022 were selected to analyze the process of water vapor
condensation on leaves, as shown in Fig. 4. It can be seen that the dew mainly appeared between 22:00 and 9:00 the next day on a daily scale. The trend of the dew amount on the three typical plants was basically the same. The condensation process generally started from about 22:00, when the intensity of condensation was greater than that of evaporation, and the dew amount was increasing with time. On August 3, 2022, the peak of condensation for the three typical plants occurred around 6:00 a.m. Tribulus condensed the most at 0.16 mm, followed by Hippophae (0.12 mm) and Elm (0.07 mm). On the remaining
two typical days, condensation peaked around 7:00 am for Tribulus and Hippophae and around 6:30 am for Elm. Thereafter, the intensity of condensation was less than that of evaporation, the rate of condensation began to decrease, and the dew amount decreased until it disappeared due to evaporation at 9:00 am. There are some differences in the dew duration of condensation in different plants. Within the three typical days, Tribulus had the longest dew duration (Table 1), with the highest percentage of 83 % (September 24, 2022), followed by Hippophae, and Elm had the shortest dew duration. Among the three typical days,
the three typical plants had the largest percentage of dew duration and the maximum dew amount on September 24, 2022.

Daily dew amount of the three typical plants was in the order of Tribulus > Hippophae > Elm as shown in Fig. 4d. Condensation and evaporation are reciprocal processes. Only when the condensation intensity at night is greater than the evaporation intensity, can dew be generated on the leaf surface. This process occupies a dominant position in the condensation cycle and lasts for a long time. As the air temperature rises after sunrise, the evaporation intensity is much higher than the condensation intensity,
and the tail of the curve at this time is steeper and more drastic than that in the earlier condensation process.





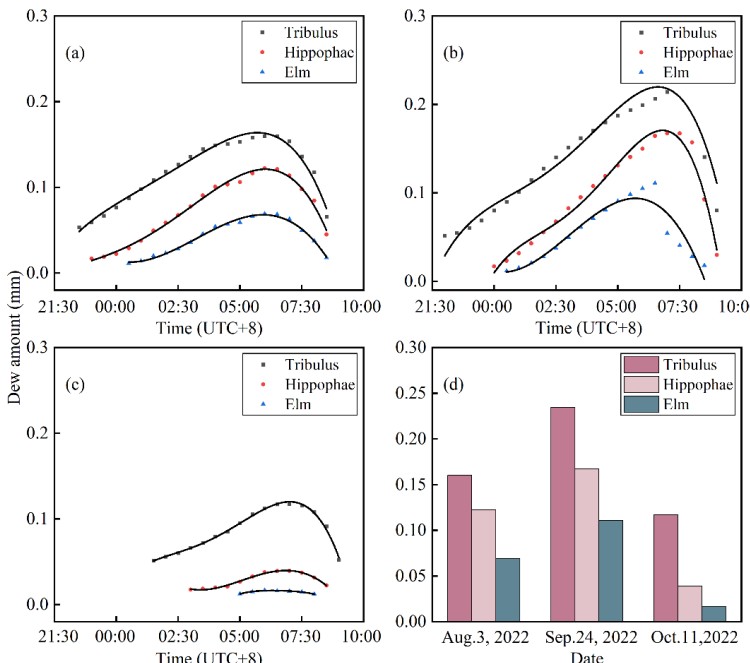

**Figure 4. Dew variations of plant leaves on August 3 (a), September 24 (b), and October 11 (c), 2022. And (d) means daily dew amount in three typical days.**

**Table 1. Dew duration in three typical plants' leaves.**

| Typical Day | Typical Plant | Dew Duration/min | Condensation Time/min | Percentage of Condensation | Dew amount/mm |
|---|---|---|---|---|---|
| August 3, 2022 | Tribulus | 619 | 462 | 75 % | 0.160 |
| | Hippophae | 588 | 423 | 72 % | 0.122 |
| | Elm | 503 | 353 | 70 % | 0.069 |
| September 24, 2022 | Tribulus | 695 | 579 | 83 % | 0.235 |
| | Hippophae | 543 | 423 | 78 % | 0.167 |
| | Elm | 522 | 373 | 71 % | 0.111 |
| October 11, 2022 | Tribulus | 457 | 336 | 74 % | 0.117 |
| | Hippophae | 363 | 257 | 71 % | 0.039 |
| | Elm | 228 | 87 | 38 % | 0.017 |





### 3.2 Monthly variation of dew on plant leaves

Monthly variation of dew on Tribulus, Hippophae and Elm leaves in May-October 2022 was shown in Fig. 5. Leaf dew occurred less frequently in May and June. 12 days occurred for Tribulus and Hippophae, with 11 days for elm in May. 13 days occurred for elm with 15 days for Tribulus and Hippophae in June. From July to October, the monthly dew days of each plant

above are more than 20 days, and those reaches 28 days in September. Dew amount was larger in September, with 5.15, 3.39, and 1.99 mm for Tribulus, Hippophae and Elm, respectively, and the smallest amount in May, with 1.08, 0.70, and 0.40 mm for the three plants above, respectively. Due to the difference in the tomentose structure of Tribulus leaf surfaces from Elm and Hippophae, the dew amount of Elm and Hippophae was less than that of Tribulus. The total dew amount in May-October 2022 was in the order of Tribulus (17.46 mm) > Hippophae (11.14 mm) > Elm (5.88 mm).


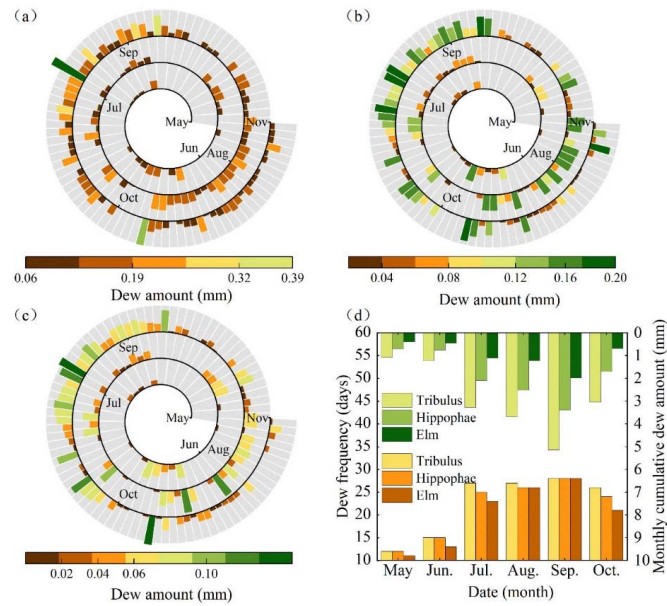

**Figure 5. Monthly variation of dew on Tribulus(a), Hippophae (b) and Elm(c) leaves. And (d) means monthly cumulative dew amount and dew frequency.**


### 3.3 Significance analysis

The results of one-way ANOVA for the daily dew on different plant leaves during May-October, 2022 were showed in Fig. 6. There were significant differences ($p<0.05$) in the amount of dew on different plant leaves. The average daily dew on Tribulus leaves was the highest at 0.140 mm, which was significantly greater than that on Hippophae leaves (0.072 mm) and that on Elm leaves (0.052 mm). The reason for the significant difference in daily dew of the three plants may be the difference in leaf inclination and that in leaf microstructure.

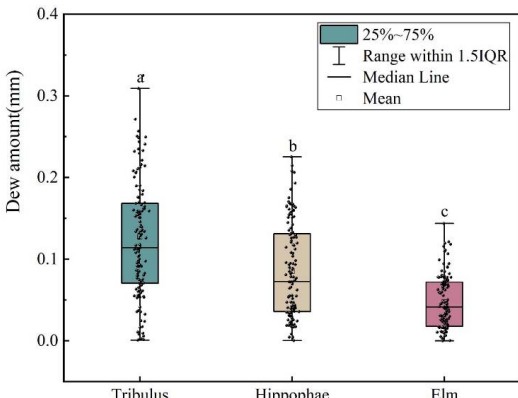

**Figure 6. Significance analysis in dew amount on different plants leaves ($p = 0.05$).**

### 3.4 Dew amount and rainfall

Monthly rainfall and monthly dew amount for Tribulus, Hippophae and Elm from May to October 2022 were shown in Table 2. The maximum value for the ratio of dew to rainfall on three plants was appeared in September, in which the proportion of Tribulus, Hippophae and Elm were 51.5 %, 33.9 % and 19.9 %, respectively. The total dew amount on Tribulus, Hippophae and Elm from May to October 2022 was 17.5, 11.1 and 5.9 mm, which was much less than the rainfall during the same period. The number of dew days was 137, 132 and 124 days, respectively, with the frequency of 74.5 %, 71.7 % and 67.4 %. The number of rain days was 98 days from May to October, 2022, with the frequency of 53.3 %, which was less than that of dew. This shows that dew can occur frequently though the dew amount is less than rainfall. The dew amount accounts for a small percentage compared with rainfall, but it can be used as an important local water recharge in the months with less rainfall.

**Table 2. Dew amount and rainfall.**

| Month | Tribulus | Hippophae | Elm |
|---|---|---|---|
| | | | |



|  | $P$/mm | $D_\mathrm{m}$/mm | $\frac{D_m}{P}$/% | $D_\mathrm{m}$/mm | $\frac{D_m}{P}$/% | $D_\mathrm{m}$/mm | $\frac{D_m}{P}$/% |
|---|---|---|---|---|---|---|---|
| May | 59.9 | 1.08 | 1.80 | 0.7 | 1.17 | 0.4 | 0.67 |
| June | 48.6 | 1.23 | 2.53 | 0.75 | 1.54 | 0.45 | 0.93 |
| July | 304.2 | 3.28 | 1.08 | 2.09 | 0.69 | 1.11 | 0.36 |
| August | 189.9 | 3.68 | 1.94 | 2.52 | 1.33 | 1.23 | 0.65 |
| September | 10.0 | 5.15 | 51.50 | 3.39 | 33.90 | 1.99 | 19.90 |
| October | 64.8 | 3.05 | 4.71 | 1.69 | 2.61 | 0.69 | 1.06 |
| Mean value | 112.9 | 2.9 | 2.58 | 1.9 | 1.64 | 1.0 | 0.87 |
| Total | 677.4 | 17.5 | 2.58 | 11.1 | 1.64 | 5.9 | 0.87 |

Ten typical rainfalls were selected to analyze the daily dew amount in the day before and after rainfall. As shown in Fig. 7, the
dew amount of the three plants after rainfall was significantly greater than that before rainfall ($p<0.01$). Among them, the average dew amount after rainfall was 3.02 times higher than that before rainfall for Tribulus, 3.28 times higher than that before rainfall for Hippophae, and 3.62 times higher than that before rainfall for Elm.

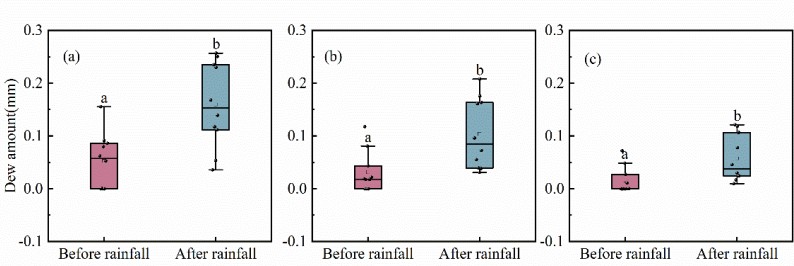

**Figure 7. Dew amount on the leaves of Tribulus(a), Hippophae(b) and Elm(c) before and after rainfall.**

**3.5 Factors affecting dew formation**

**3.5.1 Wind speed**

The correlation between the wind speed and dew amount of three typical plant leaves was shown in Fig. 8. It can be seen that the peak of dew amount corresponds to a wind speed around 0.5 m/s. When the wind speed was less than 1 m/s, the dew
frequency was high. When the wind speed was greater than 2 m/s, almost no dew occurred. Overall, as shown in Fig. 8d, the averaged dew amount increased with the increase of wind speed when the wind speed was less than 0.5 m/s. The averaged dew amount decreased with the increase of wind speed when the wind speed was more than 0.5 m/s.

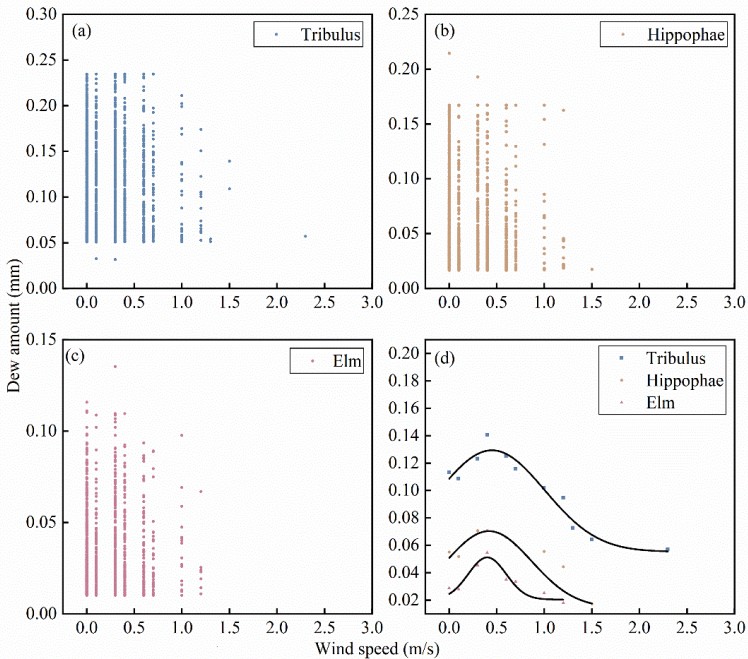

**Figure 8. The correlation between wind speed and leaf dew of Tribulus(a), Hippophae(b) and Elm(c). And (d) means dew amount**
**at each wind speed for three plants.**

### 3.5.2 Wind direction

The correlation between the wind direction and dew amount of three typical plant leaves was shown in Fig. 9. The frequency of dew on the three plants was highest at wind directions from 290° to 330°, i.e. 42 % for Tribulus, 43 % for Hippophae and 45 % for Elm. However, wind direction had less effect on the dew amount of the three plants.



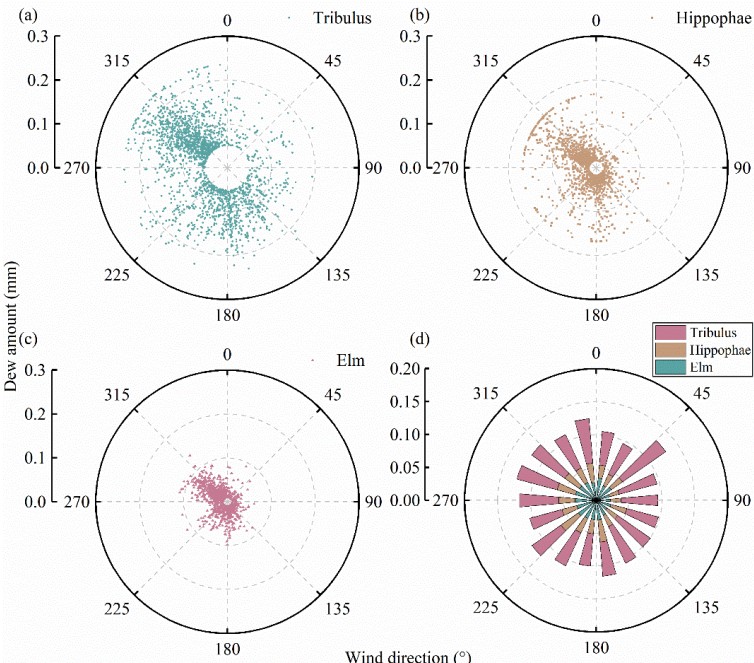


**Figure 9. The correlation between wind direction and leaf dew of Tribulus(a), Hippophae(b) and Elm(c). And (d) means dew amount at each wind direction for three plants.**

### 3.5.3 Relative humidity

The relationship between dew amount and relative humidity on the three plants was shown in Fig. 10. Dew was positively

correlated with relative humidity, and dew amount and dew frequency increased with the increase of relative humidity. Dew on the leaves of the three plants mainly occurred when relative humidity was greater than 80.0 %, and the averaged dew amount peaked when the relative humidity was more than 90.0 %, i.e. 0.13 mm for Tribulus, 0.07 mm for Hippophae, 0.03 mm for Elm.





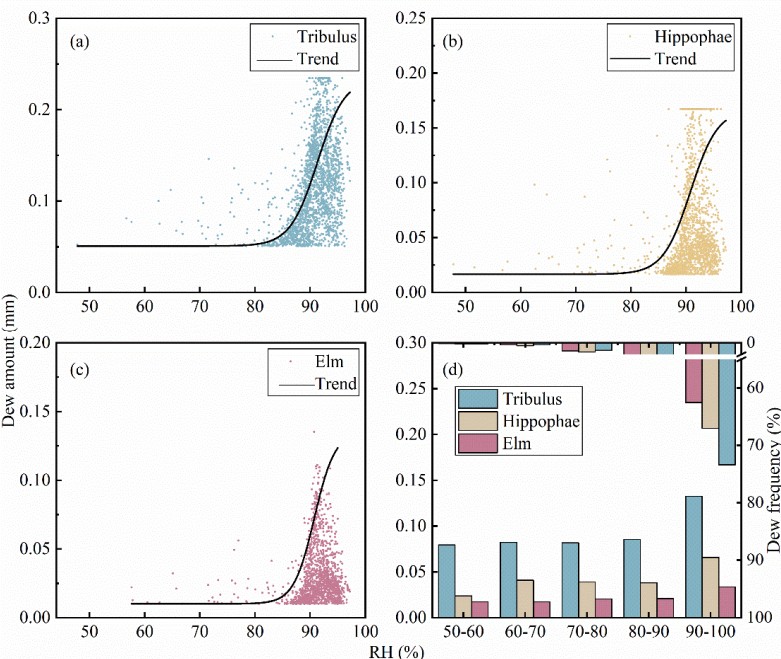

**Figure 10. The correlation between RH and leaf dew of Tribulus (a), Hippophae (b) and Elm (c). And (d) means averaged dew amount and dew frequency with the RH for three plants.**

### 3.5.4 Temperature-dewpoint difference

The amount and frequency of dew on the three plants under different temperature-dewpoint differences were shown in Fig. 11, and dew amount decreased with the increase of temperature-dew point difference. When the difference between air temperature and dew point was less than 2°C, the frequency and averaged amount of dew on the leaves of Tribulus, Hippophae and Elm were the highest, with the dew frequency of 84 %, 89 % and 93 %, and the dew amount of 0.25 mm, 0.11 mm and 0.06 mm, respectively. whereas, when the difference between air temperature and dew point was greater than 3°C, the frequency of dew on the leaves of the three plants was the lowest, with the frequency of 3 %, 3 %, and 2 %.





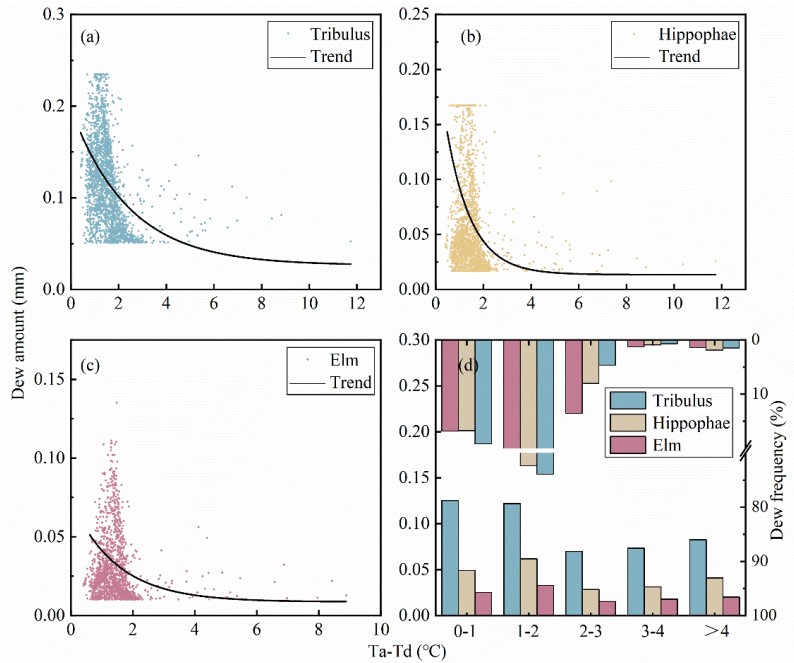

**Figure 11. The correlation between temperature-dewpoint difference and leaf dew of Tribulus (a), Hippophae (b) and Elm (c). And (d) means averaged dew amount and dew frequency with the temperature-dewpoint difference for three plants.**

**4 Discussion**

Daily variation of dew on the leaves of the three plants showed that dew mainly appeared from 22:00 to 9:00 the next day on a daily scale, and dew amount was larger before and after sunrise (5:00-7:00 am). This was in agreement with the results at the southern edge of the Mu Us Desert in China (Guo et al., 2022). In three typical days, the relationship between the dew amount of three typical plants was the largest for Tribulus, followed by Hippophae, and the smallest for Elm. Condensation and evaporation were a mutually reversible process. Dew occurred when the condensation intensity was greater than evaporation intensity at night. This process lasted for a long time. As the temperature rises around sunrise, the evaporation intensity was greater than the condensation intensity, and it was difficult for the water vapor to condense, at which time the tail of dew amount curve was steeper and more drastic compared to the pre-condensation process (Pan et al., 2010; Xu et al., 2022).

Leaf dew of Tribulus, Hippophae and Elm occurred less frequently in May and June, and dew amount was the smallest in May; monthly dew days of each leaf in July-October was more than 20 days, of which the number of dew days reached 28 days in





September and the cumulative dew amount reached the maximum. This was consistent with the results of other scholars (Scherm and van Bruggen, 1993; Hao et al., 2012; Zhang et al., 2015). The main reason was that the temperature difference between day and night was large in autumn, when the air temperature reduced at night, and the water vapor reached an appropriate range, dew occurred. On the contrary, the solar radiation was intense in Summer, coupled with the temperature rise, evaporation speed is faster, it was not easy to condensate (Clus et al., 2009; Kidron, 2010; Wang and Zhang, 2011).

Among the factors influencing dew generation, relative humidity and temperature-dewpoint difference were the main factors controlling dew generation (Sharan et al., 2007). Relative humidity was the "endogenous factor" that provided the water vapor required for dew formation. Air temperature provided "driving force" for dew formation. Monteith (1957) noted requires only that the temperature of the dew surface be below the ambient dew-point temperature, and does not require a relative humidity of 100 %. In this study, dew on the leaves of the three plants occurred mainly at RH greater than 80.0 %. The frequency of

dew on the leaves of the three plants was low when the relative humidity was less than 70 %, which was consistent with the results of Guo et al (2016). In this paper, the frequency and averaged amount of dew on the leaves of Tribulus, Hippophae and Elm were the high when the difference between the air temperature and the dewpoint was less than 2°C, with dew frequency of 84 %, 89 % and 93 %, and dew amount of 0.25, 0.11, and 0.06 mm respectively. The process of rainfall will be accompanied by the change of air temperature and relative humidity, and present studies found that the dew amount on the leaves of the

three plants was significantly greater after rainfall than before rainfall ($p < 0.01$). This was due to the increase in air humidity caused by water vapor transport before rainfall, and the changes in air humidity and temperature caused by evapotranspiration from plants and soil after rainfall, which had a direct effect on the formation of dew. Especially in arid areas, the formation process of dew had a very obvious relationship with the distribution of precipitation. At the same time, the surface of plants or other objects on the ground after rainfall was more humid, which is more conducive to condensation (Fang, 2020; Yu et al.,

285 2023).

Wind speed affect water vapor flow and energy transfer in the atmosphere. Too much or too little wind speed can limit the formation of dew, and moderate wind speeds allow water vapor to be replenished (Zhang et al., 2015), which is conducive to increasing the relative humidity of the air. In this study, it was found that dew formation was favored when the wind was less than 1.0 m/s, as opposed to when the wind speed was greater than 2 m/s, almost no dew was produced. Monteith's study (1957)

concluded that dew formation requires a wind speed of <0.5 m/s; while Muselli et al (2002) concluded that a wind speed of <1.0 m/s can provide sufficient moisture for the formation of dew. It had also been concluded that the critical line for wind speeds favor to dew was 4.5 m/s (Beysens et al., 2005). Obviously, a certain wind speed can promote water vapor transport and be favor to dew formation (Vuollekoski et al., 2015; Zhang et al., 2015; Beysens, 2016; Zhuang and Zhao, 2017).

In this study, dew mainly occurred in the wind direction of 290° - 330°, which is the main wind direction in the region. So it

will inevitably bring more water vapor sources to promote dew production. This was inconsistent with the findings of Sharana et al. (2007) in Kothara region, India, which may be due to the fact that the role of wind direction on dew varies depending on the regional geographic location, the main regional wind direction and the source of water vapor, so the frequency distribution of dew with the wind direction in the present study are only applicable to study region.



There were significant differences in the dew amount on the leaves of Tribulus, Hippophae and Elm. The dew amount on
Tribulus was significantly greater than that on Hippophae, and the dew amount on Elm was the least. The reason for the
difference may be the difference in leaf traits of the three plants. The morphology and internal structure of the plant itself will
affect the absorption of dew (Rundel, 1982; Holanda et al., 2019). Irregular leaf surface with peridiums can capture small water
droplets. Tribulus leaves had dense and fine tomentum, this structure increased the roughness and surface area of the leaf, thus
allowing a larger area for dew to collect and deposit on leaf surface, increasing the dew amount on leaf surface. The structure
of the tomentum reduced evaporation on leaf surface, allowing the temperature of plant leaf to decrease, thus increasing dew
condensation and aggregation. The structure of the tomentum also prevent droplets from slipping off the leaf surface, thus
increasing the residence time of dew on leaf surface. This resulted in a significantly higher dew amount on the Tribulus leaves
than that on Hippophae and Elm leaves. In addition, the leaf inclination angle may also have an effect on the dew amount.
Previous studies have shown that there was a difference in the amount and duration of dew under different leaf inclination
angles, and the dew amount decreased with increasing inclination angle (Sentelhas et al., 2004; Kidron, 2005). The main reason
was that the effective radiant area decreased with the increase of inclination angle, which changed the sky view factor (SVF)
and ultimately had an effect on the temperature of the condensation surface. According to field observations, the leaves of
Tribulus were close to the surface, almost level with the ground, and were less disturbed by wind speed and other factors,
while the leaves of Hippophae and Elm were higher from the ground, and the leaf inclination was constantly changing under
the influence of gravity and wind speed. This may be one of the reasons for the differences in the amount of dew of Tribulus,
Hippophae and Elm leaves.

In this study, it was found that the dew amount of Tribulus, Hippophae and Elm was higher on the day after rainfall than that
before rainfall, which was consistent with the results in the Loess Plateau area by Zhang Qiang et al (2015), The main reason
was that the water vapor condition in the air after rainfall was sufficient (Yokoyama et al., 2021), and the weather after rain
was mostly sunny and less cloudy, which was conducive to radiation cooling at night, and thus conducive to the dew generation.
Dew was mostly condensed on the leaves of crops and natural vegetation. Dew amount per unit area of land was mainly
influenced by dew intensity and leaf area index (LAI). Dew amount on the leaves of plants in this study was the dew intensity,
which will further assess regional dew in combination with the LAI.

## 5 Conclusions

Three typical plants, namely Tribulus, Hippophae and Elm, were selected in the loess hilly region of China. The characteristics
of dew on plant leaves were analyzed by manual measuring the leaf dew and combining with the automatic observation data
by LWS. It can be concluded that:

(1) Dew amount of typical plant leaves from May to October 2022 was in the order of Tribulus (17.46 mm) > Hippophae
(11.14 mm) > Elm (5.88 mm). The difference of dew amount was mainly related to the leaf inclination angle and the
microstructure of the leaves.



(2) Dew on plant leaves mainly appeared between 22:00 and 9:00 the next day and was mainly concentrated in July-October during the year. Dew amount on leaves was smaller than the amount of rainfall, but the frequency of dew was higher than the frequency of rainfall.

(3) Dew was more likely to occur when the relative humidity was >80.0 %, the temperature-dew point difference was less than 335    2 °C and the wind speed was less than 1.0 m/s, and dew amount was also significantly increased after rainfall.

**Data availability**

Data will be made available on request.

**Author contributions**

**Zhifeng Jia:** Conceptualization, Methodology, Supervision, Funding acquisition, Project administration, Data curation, 340    Formal analysis, Investigation, Resources, Writing – review & editing. **Ge Li:** Methodology, Investigation, Data curation, Formal analysis, Writing - Original Draft. **Cheng Jin:** Data curation, Investigation. **Jun Xing:** Data curation, Methodology. **Yingjie Chang:** Conceptualization, Methodology, Formal analysis, Writing-review & editing. **Pengcheng Liu:** Formal analysis, Data curation, Supervision. **Danzi Chen:** Formal analysis, Supervision.

**Competing interests**

The contact author has declared that none of the authors has any competing interests.

**Acknowledgments**

This work was supported by the Fundamental Research Funds for the Central Universities, CHD (300102293209), the National Natural Science Foundation of China (42001033) and the Natural Science Basic Research Plan in the Shaanxi Province of China (2021JQ-237).

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
