# Peer review of "Characteristics of dew on leaves of typical plants in Loess Hill and Gully Region of China"

_Hydrology and Earth System Sciences, 2024_

## Referee Comment (RC2)

**Review of HESS-2024-94**

Characteristics of dew on leaves of typical plants in Loess Hill and Gully Region of China

General comments

The authors studied dew formation in three different leaf types in a semi-arid region of China for a year, using automatic meteorological measurements from a station and analyzing leaf samples weighted in the lab. Even though the authors intend to explain the differences in dew measurements using meteorological data, they must provide sufficient explanation of these differences beyond meteorological correlations, which are very low.

My main concerns of the papers are:

The authors base their measurements on a leaf wetness sensor (LWS) and leave samples that are weighed in the laboratory. The leaf wetness sensor does not differentiate between fog and dew nor the water collected by the leaves. The authors should clearly prove that what leaves or LWS is collecting is dew and not fog, which is a different atmospheric water non-rainfall input. My concern arises since the main explanation of the dew amount measured by the authors is based on the difference between air temperature and dew temperature < 2C, which would mean fog and not dew. The authors should prove that fog does not play any role in the water that is being collected.

Section 3.5 requires more than just simple descriptions of meteorological conditions, this mainly because authors make correlations, which are apparently very low and any statistical parameter is included ($r^2$, r, slope, etc). Authors such as Lobos-Roco et al., 2024 and de Roode et al., 2010, have shown that dew occurs under specific atmospheric conditions, including a stable boundary layer, low wind speed, and a negative specific humidity tendency (dq/dt) of around -1.5 g/kg/h. These more detailed variables could provide a better understanding of dew formation than simple descriptions of relative humidity, wind speed, and wind direction. Therefore, it is essential for the authors to include these detailed variables in their analysis to enhance the comprehensiveness of their study.

The introduction of the manuscript could benefit from being more extensive and detailed. It should include a comprehensive description of the study area, the origin of the moisture that produces dew, the synoptic atmospheric processes that provide moisture/rain to the region, and a clear definition of the processes involved in dew formation. This extended introduction would provide a clearer context for the study, helping the readers to better understand the significance and relevance of the research.

The English of the entire manuscript must be checked. The whole manuscript is written in past tense. For example, 'was shown in Figure ..' It should be 'is shown'.

Specific comments

*Title:*

Title must be revised, it is too much similar to https://doi.org/10.3390/w11010126, 2019.

*Abstract:*

Line 14: I suggest to use the term of "non-rainfall", because dew is also a kind of precipitable water, water vapor downward flux.

*Main body:*

Line 31: Please add a more readable number, like % of arid-semi-arid compare to total China Area, 30%? 20%?

Line 53: what do you mean by "typical"?

Line 60: An international reader shouldn't know where the experimental station is by the address. Please provide coordinates, altitude and other geographical characteristic to help the reader to understand the site.

Line 63: to refer to altitude (no height) you need to add the meters above sea level (m ASL).

Line 64: what is "d" after 157? please clarify.

Fig 1: Remove de minutes and seconds in the coordinate systems if they are not providing any additional information. Please attached a small map of China, to see where the place is located. I suggest to change the color of 'DEM' to a different palette color that the one of NDVI. Delete the word "county" fro the label in the maps, if it is clearly stated in the legend.

Line 100: What do you mean by "manual"? please elaborate more.

Line 111: introduce the equation

Line 122: Wd was already defined in equation 1

Line 137: I think that dew is not affected by meteorological 'factors' but by meteorological 'conditions'.

Line 154: Why only three days if the total measurement period was higher than a year? Why don't you use the entire data period showing means + standard deviations?

Line 177: Please use 'present' verb form. It is shown, not "was" shown.

Figure 5: Even though this spiral plot are a fancy way to show the monthly cycle, it is quite hard to read them. It is not clear the monthly cycle. I suggest to use monthly averages in a classic barplot. Color bar is not the same for every plant-type, please standarize. (d) Please indicate if green is frequency or dew amount, same for brown. It is not clear.

Line 190-195: Authors stated that significant differences in dew amount between the different leaves is because the microstructure and leave inclination. However, they do not provide any prove of that. Please if you state that inclination and mircostructure of leaves is the main characteristic that differentiate the dew amount, at least describe it.

Figure 6: the mean (black square) is hidden behind the dots. Please make it bigger. What are the dots?

Table 2: Explain what does it mean the columns name. For example: "Dm/mm".

Line 216: I think they are not "factors" the are "atmospheric conditions"

Line 218-222: The paragraph seems to be contradictory. Dew frequency decreases with high wind speed but this wind speed increases the dew amount? How can be possible that dew frequency increases and dew amount decreases?

Figure 8: I do not see any correlation between wind speed and dew amount. It need more explanation of why correlated wind speed when it apparently does not play any direct role on dew. R2 and curve slope are not included. Authors should find another statistical parameters to relate wind speed and dew.

3.5.2 Wind direction: I think that analyzing wind direction does not make any sense since wind speed is extremely low. When wind speed is lower than 2 m/s, wind direction is random. Instead of wind

speed and direction, authors must include measurements of atmospheric turbulence, which are more related to dew formation (stable boundary layer)

Line 245: Here is main main issue of the paper. Dew is a surface processes resulted from the surface radiative loss. This radiative loss cool down the surface temperature close or below the dew point, condensing atmospheric water vapor over the surface. If air temperature is close the dew point, it does not mean that dew is present. It means that air is condensing, resulting in fog. Then, it is not possible to relate dew formation over different leaves surfaces using air temperature. The authors should use leave surface temperature instead of air temperature.

Line 295: water vapor from where?

Line 300 to 315: I think authors should include data of the leave roughness and inclination angle in the results section to after discuss here they hypothesis. Note that leave roughness also can be favourable to collect fog.

Line 400: Consider to change the title, which is very similar to the one in line 400.

Suggested literature to be checked:

- Ritter, F., Berkelhammer, M., & Beysens, D. (2019). Dew frequency across the US from a network of in situ radiometers. Hydrology and Earth System Sciences, 23(2), 1179-1197.

- de Roode, S. R., Bosveld, F. C., & Kroon, P. S. (2010). Dew formation, eddy-correlation latent heat fluxes, and the surface energy imbalance at Cabauw during stable conditions. Boundary-layer meteorology, 135, 369-383.

- Lobos-Roco, F., Suárez, F., Aguirre-Correa, F., Keim, K., Aguirre, I., Vargas, C., Abarca, F., Ramírez, C., Escobar, R., Osses, P., et al. (2024). Understanding inland fog and dew dynamics for assessing potential non-rainfall water use in the Atacama, Journal of Arid Environments, 221, 105 125, 2024.

---

## Author Comment (AC1)

**Response to Anonymous Referee #1 Comments**

We feel great thanks for your professional review work on our article. As you are concerned, there are several problems that need to be addressed. According to your nice suggestions, we have responded. The details of the reply are as follows:

**Point 1:** Introduction section: In this section, the authors need to specify the scientific problems and highlight the innovations and academic contributions of the study.

**Response:** We fully agree with your suggestion that it helps to enrich the content and increase the quality of the manuscript. We will revise the introduction section. The scientific problem will be clarified in the research background section of the introduction, and the innovation of the study and the ecological significance of studying dew will be highlighted in the research objectives section of the introduction.

**Point 2:** Figure 1: The line of "Provincial boundary" should be the boundary of Yan'an City, but not the one of Shaanxi Province.

**Response:** We are very grateful to you for pointing out the details in the manuscript. We found this error in the manuscript, and in the next revision we will correct the "provincial boundary" in Figure 1 to read "Yan'an city boundary". The modified Figure 1 is as follows:

[Figure]

**Figure 1. Location of the experimental site (Jia et al., 2023): (a) China map, (b) Geographic location of the loess hill and gully area, (c) Geographic location of the study area.**

**Point 3:** Section 2.2: Please give out the field pictures of the three plants of Tribulus, Hippophae, and

Elm from different angles. It is better to include their pictures in different growth period.

**Response:** Thank you for your comment. Our artificial field observation experiments were conducted in September-October 2021 and September-October 2022. No pictures in other periods were taken because leaf collection was not carried out during the other growth periods of the plants. Figure 2 shows images of plant leaves taken during the artificial field observation period, which we will add to the manuscript.

[Figure]

**Figure 2. The field pictures of the three plants. (a) Tribulus(19 October 2022); (b) Hippophae(16 October 2022); (c) Elm(16 October 2022); (d) Tribulus leaves sampling(18 September 2022); (e) Hippophae leaves sampling(19 September 2022); (f) Elm leaves sampling(19 September 2022).**

**Point 4:** Section 2.3: Why did the authors place the sensors/equipments of ATMOS14, WSD01, and ECRN-100 at 0.2 m, 1.0 m, and 1.6 m, respectively above the ground? Are there any standards to do that?

**Response:** We are very grateful to your valuable comment. There is no prescribed standard for the mounting height of the sensors/devices of ATMOS14, WSD01 and ECRN-100, and we mainly referred

to the existing studies in the experimental design: the mounting height of the ATMOS14 sensor was referred to the mounting height of the air temperature and relative humidity sensors in the article of Jia et al (2023); the mounting height of the WSD01 anemometer was referenced to the mounting height of anemometer in the article Zhang et al (2015); the mounting height of ECRN-100 rain gauge was referenced to the mounting height of ECRN-100 high-resolution rain gauge in Jia et al (2023);

**Point 5:** Figure 2(a): Please give out the date/time when you took the photo.

**Response:** We gratefully appreciate your comment. The date of Fig. 2(a) was taken on 18 May 2022 at 7:32 a.m. The original image is shown in Fig. 3.

[Figure]

**Figure 3. The original image of Fig. 2(a)**

**Point 6:** Line 103: Why are the number (5 and 10) different for these three plants? Did you collect leaves everyday?

**Response:** We gratefully appreciate your comment. In the artificial field observation experiment, we measured the leaf area of Tribulus, Hippophae and Elm with the SYS-Leaf1000 leaf image analyzer, respectively. The average leaf area was 320.40 mm$^2$ for Tribulus, 187.66 mm$^2$ for Hippophae, and 398.94 mm$^2$ for Elm. Because the leaf area of Hippophae was much smaller than that of Tribulus and Elm, and in order to better observe the variation of dew amount on the leaves of the plants, five leaves were taken from the leaves of Tribulus and Elm and ten leaves from the leaves of Hippophae when taking samples. Plant leaf shape characteristics show in Table 1.

**Table 1. Plant leaf shape characteristics**

| Type | Leaf | Maximum | Minimum | Median | Standard | Mean | Coefficient |
|------|------|---------|---------|--------|----------|------|-------------|

| | Characters | | | | deviation | | of Variation |
|---|---|---|---|---|---|---|---|
| Tribulus | Area (mm²) | 560.44 | 35.20 | 295.45 | 89.75 | 320.40 | 0.28 |
| | Aspect Ratio | 3.91 | 0.40 | 1.11 | 0.42 | 1.03 | 0.41 |
| | Perimeter (mm) | 367.96 | 25.36 | 211.28 | 41.90 | 215.14 | 0.19 |
| | Leaf Shape Coefficient | 0.69 | 0.05 | 0.09 | 0.06 | 0.10 | 0.57 |
| Hippophae | Area (mm²) | 301.45 | 83.24 | 191.97 | 49.58 | 187.66 | 0.26 |
| | Aspect Ratio | 4.69 | 0.18 | 3.16 | 1.51 | 2.44 | 0.62 |
| | Perimeter (mm) | 117.56 | 46.70 | 77.44 | 13.76 | 76.22 | 0.18 |
| | Leaf Shape Coefficient | 0.55 | 0.26 | 0.43 | 0.06 | 0.42 | 0.14 |
| Elm | Area (mm²) | 866.56 | 166.84 | 381.75 | 140.48 | 398.94 | 0.35 |
| | Aspect Ratio | 2.28 | 0.45 | 1.57 | 0.60 | 1.27 | 0.47 |
| | Perimeter (mm) | 146.07 | 51.01 | 88.20 | 18.20 | 89.93 | 0.20 |
| | Leaf Shape Coefficient | 0.78 | 0.47 | 0.60 | 0.06 | 0.61 | 0.10 |

**Point 7:** Equation (1): Please give out the unit of $S_i$ in the explanation section of the equation.

**Response:** We are very grateful to you for pointing out the details in the manuscript. In Eq. (1), $S_i$ is the area of the ith plant leaf in cm². In manuscript revisions, we will explain this in the explanation section of the equation.

**Point 8:** Equation (4): I feel that this equation is incorrect. For example, when $W_{dmax}=W_{dmin}$ and both are not zero, the result $D_d=0$ is obviously unrealistic.

**Response:** We are very grateful to you for pointing out the details in the manuscript. We agree with you. We found an ambiguity in this equation in the manuscript. In our revision, we will correct this equation by replacing $W_{dmax}$ with $W_{d_i max}$ and $W_{dmin}$ with $W_{d_i min}$, where $W_{d_i max}$ is the maximum dew amount monitored in the $i$th period (mm); and $W_{d_i min}$ is the minimum dew amount monitored in the ith period (mm). The revised Eq. (4) is:

$$D_d = \sum_{i=1}^{n} (W_{d_i max} - W_{d_i min}), \tag{4}$$

**Point 9:** Section 3.5: Are there other factors affecting dew formation? In Section 2.2, it is apparent that the heights of these three types of plants differ. When collecting leaves, are the heights of the leaves from the ground the same? Does height also affect the amount of dew?

**Response:** We are very grateful to your valuable comment. It has been shown that meteorological factors and substrate characteristics are the most important factors affecting dew formation (Pan et al., 2010), with lower air temperatures (Monteith, 1956; Li, 2002), higher air relative humidity (Zangvil, 1996; Ye et al., 2007), and moderate wind speeds (Zhang et al. 2015; Beysens, 2016; Zhuang and Zhao, 2017) are the most suitable meteorological conditions for dew formation. In addition, differences in subsurface properties such as soil texture, soil moisture content, and surface roughness can affect dew formation in different regions (Tao and Zhang, 2012). Since the object of our present study is plant leaf condensation rather than soil condensation, only the effect of meteorological factors on dew formation is considered.

When we collected plant leaves, the collection heights of leaves of the same species were basically the same, i.e. 0.2m for Tribulus leaves, 0.6m for Hippophae leaves and 1.5m for Elm leaves.

Different heights can have an effect on the amount of dew on the leaves of different plants, and the height factor was weakened during sampling by incorporating the growth characteristics of different plants, which we have described in the discussion.

**Point 10:** Section 3.5.1: The installation height of the WSD01 anemometer is fixed (1 meter above the ground), which is approximately the same as the height of the Hippophae, but differs from the heights of other trees. Therefore, is it appropriate to quantify the formation of dew for these three types of trees using the same wind speed?

**Response:** We are very grateful to your valuable comments, which helped to improve the readability of the manuscript. The wind speeds at different heights can be converted according to the formula, which is as follows (Monteith and Unsworth, 1990; Sharan et al 2007). We will analyze the influence factors of the three plants separately according to the formula converted to wind speed at different heights.

$$\frac{v_{z_1}}{v_{z_2}} = \frac{\ln\left(\frac{z_1}{z_C}\right)}{\ln\left(\frac{z_2}{z_C}\right)} \tag{1}$$

where $Z_c$ (m) is the roughness length and taken as equal to 0.1 m, $V_{Z_1}$ (m/s) and $V_{Z_2}$ (m/s) are wind speeds at different heights of $Z_1$ (m) and $Z_2$ (m), respectively.

**Point 11:** The manuscript only considers the amount of dew in three typical days, which is not representative enough. More daily calculation results should be added.

**Response:** We fully agree with your suggestion that it helps to enrich the content and increase the quality of the manuscript. To make the study more rigorous, we will add the average daily dynamics of plant leaf dew in May-October 2022 to the revision process

**References**

Beysens, D.: Estimating dew yield worldwide from a few meteo data, Atmos. Res., 167, 146-155, https://doi.org/10.1016/j.atmosres.2015.07.018, 2016.

Jia, Z. F., Wu, B., Wei, W., Chang, Y., Lei, R., Hu, W., and Jiang, J.: Effect of Plastic Membrane and Geotextile Cloth Mulching on Soil Moisture and Spring Maize Growth in the Loess–Hilly Region of Yan'an, China, Agronomy, 13, https://doi.org/10.3390/agronomy13102513, 2023.

Li, X. Y.: Effects of gravel and sand mulches on dew deposition in the semiarid region of China, J. Hydrol., 260, 151-160, https://doi.org/https://doi.org/10.1016/S0022-1694(01)00605-9, 2002.

Monteith, J. L.: Dew, Q. J. R. Meteorol. Soc., 83, 322-341, https://doi.org/10.1002/qj.49708335706, 1957.

Monteith, J. L. and Unsworth, M.H.:  Principles of Environmental Physics, second ed. Routledge, Chapman &Hall, Inc., New York,1990.

Pan, Y. X., Wang, X. P., and Zhang, Y. F.: Dew formation characteristics in a revegetation-stabilized desert ecosystem in Shapotou area, Northern China, J. Hydrol., 387, 265-272, https://doi.org/10.1016/j.jhydrol.2010.04.016, 2010.

Sharan, G., Beysens, D., and Milimouk-Melnytchouk, I.: A study of dew water yields on Galvanized iron roofs in Kothara (North-West India), J. Arid. Environ., 69, 259-269, https://doi.org/10.1016/j.jaridenv.2006.09.004, 2007.

Tao, Y. and Zhang, Y. M.: Effects of leaf hair points of a desert moss on water retention and dew formation: implications for desiccation tolerance, J. Plant Res., 125, 351-360, https://doi.org/10.1007/s10265-011-0449-3, 2012.

Ye, Y. H., Zhou, K., Song, L. Y., Jin, J. H., and Peng, S. L.: Dew amounts and its correlations with meteorological factors in urban landscapes of Guangzhou, China, Atmos. Res., 86, 21-29, https://doi.org/https://doi.org/10.1016/j.atmosres.2007.03.001, 2007.

Zangvil, A.: Six years of dew observations in the Negev Desert, Israel, J. Arid. Environ., 32, 361-371, https://doi.org/https://doi.org/10.1006/jare.1996.0030, 1996.

Zhang, Q., Wang, S., Yang, F. L., Yue, P., Yao, T., and Wang, W. Y.: Characteristics of dew formation and distribution, and its contribution to the surface water budget in a Semi-arid Region in China, Bound.-Layer Meteor., 154, 317-331, https://doi.org/10.1007/s10546-014-9971-x, 2015.

Zhuang, Y. L. and Zhao, W. Z.: Dew formation and its variation in Haloxylon ammodendron plantations at the edge of a desert oasis, northwestern China, Agric. For. Meteorol., 247, 541-550, https://doi.org/10.1016/j.agrformet.2017.08.032, 2017.

---

## Author Comment (AC2)

**Response to Anonymous Referee #2 Comments**

**General comments:** The authors studied dew formation in three different leaf types in a semi-arid region of China for a year, using automatic meteorological measurements from a station and analyzing leaf samples weighted in the lab. Even though the authors intend to explain the differences in dew measurements using meteorological data, they must provide sufficient explanation of these differences beyond meteorological correlations, which are very low.

**Response:** We feel great thanks for your professional review work on our article. As you are concerned, there are several problems that need to be addressed. According to your nice suggestions, we have responded. The details of the reply are as follows:

**Responses to Main Concerns:**

**Point 1:** The authors base their measurements on a leaf wetness sensor (LWS) and leave samples that are weighed in the laboratory. The leaf wetness sensor does not differentiate between fog and dew nor the water collected by the leaves. The authors should clearly prove that what leaves or LWS is collecting is dew and not fog, which is a different atmospheric water non-rainfall input. My concern arises since the main explanation of the dew amount measured by the authors is based on the difference between air temperature and dew temperature $< 2\,℃$, which would mean fog and not dew. The authors should prove that fog does not play any role in the water that is being collected.

**Response:** We are very grateful to your valuable comment. We fully agree with your suggestion. In Zhang's study (Zhang et al., 2019), a distinction was made between non-rainfall water, which is fog when the relative humidity is equal to 100 %. In our study, we excluded data with a relative humidity of 100 per cent as a way of distinguishing between dew and fog.

**Point 2:** Section 3.5 requires more than just simple descriptions of meteorological conditions, this mainly because authors make correlations, which are apparently very low and any statistical parameter is included ($r^2$, r, slope, etc). Authors such as Lobos-Roco et al., 2024 and de Roode et al., 2010, have shown that dew occurs under specific atmospheric conditions, including a stable boundary layer, low wind speed, and a negative specific humidity tendency (dq/dt) of around -1.5 g/kg/h. These more detailed variables could provide a better understanding of dew formation than simple descriptions of

relative humidity, wind speed, and wind direction. Therefore, it is essential for the authors to include these detailed variables in their analysis to enhance the comprehensiveness of their study.

**Response:** We are very grateful for your comments, which have made our manuscript more rigorous. We will analyze the factors affecting leaf dew production by including statistical parameter and performing correlation analyses in section 3.5 of the manuscript.

We agree that dew occurs under specific atmospheric conditions, but due to experimental conditions, we did not make measurements of atmospheric turbulence, so we were unable to perform boundary layer analyses. For analyses of wind speed we will include the relevant statistical parameters in the manuscript. Specific humidity and relative humidity are both indicators describing the water vapour content of the air, so we analyze relative humidity this time. We will then proceed to conduct studies or experiments on the stable boundary layer, specific humidity, etc. for more in-depth analyses.

**Point 3:** The introduction of the manuscript could benefit from being more extensive and detailed. It should include a comprehensive description of the study area, the origin of the moisture that produces dew, the synoptic atmospheric processes that provide moisture/rain to the region, and a clear definition of the processes involved in dew formation. This extended introduction would provide a clearer context for the study, helping the readers to better understand the significance and relevance of the research.

**Response:** We are very grateful to your valuable comment. We will reconstruct the introduction to the manuscript. In the introduction, add a full description of the study area, the origin of the moisture that produces dew, a clear definition of the process of dew formation, etc.

**Point 4:** The English of the entire manuscript must be checked. The whole manuscript is written in past tense. For example, 'was shown in Figure ..' It should be 'is shown'.

**Response:** We are very grateful to you for pointing out the details in the manuscript. We will check the English throughout the manuscript and make corrections based on your suggestions.

**Responses to Specific Comments:**

**Point 1:** Title must be revised, it is too much similar to https://doi.org/10.3390/w11010126, 2019.

**Response:** We are very grateful to your valuable comment. We take this issue you raised very seriously and we will make changes to the title.

**Point 2:** Line 14: I suggest to use the term of "non-rainfall", because dew is also a kind of precipitable water, water vapor downward flux.

**Response:** We are very grateful to you for pointing out the details in the manuscript. We fully agree with your suggestion. We have created an ambiguity in the expression, which we will change to "non-rainfall".

**Point 3:** Line 31: Please add a more readable number, like % of arid-semi-arid compare to total China Area, 30%? 20%

**Response:** We are very grateful for your comments, which have made our manuscript more rigorous. China's arid and semi-arid regions cover an area of more than $2.56 \times 10^6$ km$^2$, about 25 % of China's land area. (Wang et al., 2021). We'll add the exact percentage to the manuscript.

**Point 4:** Line 53: what do you mean by "typical"?

**Response:** We are very grateful to your valuable comment. "Typical" means a typical observation site that reflects the climate of the Loess Hills and Gullies region of China.

**Point 5:** Line 60: An international reader shouldn't know where the experimental station is by the address. Please provide coordinates, altitude and other geographical characteristic to help the reader to understand the site.

**Response:** We appreciate you pointing out the details of the manuscript. I will provide coordinates, elevations, and other geographic features in the manuscript to help international readers understand the site.

**Point 6:** Line 63: to refer to altitude (no height) you need to add the meters above sea level (m ASL).

**Response:** We gratefully appreciate your comment. We will standardise the units of elevation in the manuscript.

**Point 7:** Line 64: what is "d" after 157? please clarify.

**Response:** We are very grateful to you for pointing out the details in the manuscript. The "d" after 157 means "days". The correct formulation should be "the frost-free period is 157 days". We'll correct it in the manuscript.

**Point 8:** Fig 1: Remove de minutes and seconds in the coordinate systems if they are not providing any additional information. Please attached a small map of China, to see where the place is located. I suggest to change the color of 'DEM' to a different palette color that the one of NDVI. Delete the word "county" fro the label in the maps, if it is clearly stated in the legend.

**Response:** We appreciate your suggestions for Fig. 1. We will remove the words "minutes" and "seconds" from the coordinate systems and the word "county" from the map labels in the manuscript. And we will attach a map of China to Fig. 1 to indicate the experimental site. Finally, we will use more different palette colours for the "DEM" and the "NDVI". The modified Figure 1 is as follows:

[Figure]

**Figure 1. Location of the experimental site (Jia et al., 2023): (a) China map, (b) Geographic location of the loess hill and gully area, (c) Geographic location of the study area.**

**Point 9:** Line 100: What do you mean by "manual"? please elaborate more.

**Response:** "Manual" means that we conducted artificial field experiments to calibrate the LWS in September-October 2021 and September-October 2022. Leaves of three plants were picked at 6:30 a.m., 7:00 a.m., 7:30 a.m., and 8:00 a.m., respectively, and brought back to the laboratory for weighing. "Manual" is used in line 100 to correspond to "automatic observations". To avoid ambiguity, I will remove the word "Manual" from the manuscript.

**Point 10:** Line 111: introduce the equation.

**Response:** Thank you for your comment. In Equation 1, "$m_1$" is the weight of the sealed bag containing the leaf dew (g), "$m_2$" is the weight of the sealed bag (g), "$m_3$" is the dry weight of the leaf (g), "$m_1$- $m_2$- $m_3$" refers to the weight of dew condensed on leaves. "$S_i$" is the area of the ith plant leaf, and "n" is the number of leaves, where Tribulus and Elm $n$=5 and Hippophae $n$=10. "$\sum_{i=1}^{n} S_i$" refers to the sum of the leaf areas of the n leaves. "$\rho$" is the density of water (1 g/cm³). $\frac{1000(m_1-m_2-m_3)}{\sum_{i=1}^{n} S_i}\rho$ is calculated as the amount of dew per unit area of leaf, that is, $W_d$ (mm).

$$W_d = \frac{1000(m_1-m_2-m_3)}{\sum_{i=1}^{n} S_i}\rho \tag{1}$$

**Point 11:** Line 122: $W_d$ was already defined in equation 1.

**Response:** Thank you for your valuable suggestion. $W_d$ in Eq. 1 refers to the amount of leaf dew per unit area calculated in manual field observation experiments, while $W_d$ in Eq. 2 refers to the amount of leaf dew per unit area converted from measurements made by leaf wetness sensors (LWS). Although both are the amount of leaf dew per unit area, have different definitions, so they need to be stated separately. We will use different letters for the definitions in the manuscript to distinguish them.

**Point 12:** Line 137: I think that dew is not affected by meteorological 'factors' but by meteorological 'conditions'.

**Response:** We gratefully appreciate your comment. We think that "fluctuation of meteorological factors" create "meteorological conditions". We would standardize the formulation of line 137 by correcting it to read "Since the dew occurrence is affected by the meteorological conditions, dew occurs only during part of the day".

**Point 13:** Line 154: Why only three days if the total measurement period was higher than a year? Why don't you use the entire data period showing means + standard deviations?

**Response:** We fully agree with your suggestions, which will help enrich the manuscript and improve its quality. In order to make the study more rigorous, we will add in the revision process the changes in the mean daily dynamics of plant leaf dew in May-October 2022 and show the standard deviations of the mean daily dynamics.

**Point 14:** Line 177: Please use 'present' verb form. It is shown, not "was" shown.

**Response:** We are very grateful to you for pointing out the details in the manuscript. We will check the English of the whole manuscript and standardise the English presentation.

**Point 15:** Figure 5: Even though this spiral plot are a fancy way to show the monthly cycle, it is quite hard to read them. It is not clear the monthly cycle. I suggest to use monthly averages in a classic barplot. Color bar is not the same for every plant-type, please standarize. (d) Please indicate if green is frequency or dew amount, same for brown. It is not clear..

**Response:** We are very grateful for your suggestions. To make it easier for readers to read Fig. 5, we will replace the spirals in the manuscript with classic bar charts of monthly averages and standardize the color bars for each plant-type.

The green color bar in Fig. 5d represents the monthly cumulative dew amount (mm) and the brown color bar represents the dew frequency (days).

**Point 16:** Line 190-195: Authors stated that significant differences in dew amount between the different leaves is because the microstructure and leave inclination. However, they do not provide any prove of that. Please if you state that inclination and mircostructure of leaves is the main characteristic that differentiate the dew amount, at least describe it.

**Response:** Thank you for your valuable suggestion. During our field experiments, we observed that the Tribulus leaves were almost horizontal, the angle between the Tribulus leaves and the ground was about 0°. However, for most leaves, the angle between the Hippophae leaves and the ground was about 30°-60°, and the angle between the Elm leaves and the ground was about 30°-45°. We will add a corresponding description in the manuscript. Figure 2 shows the field photograph of the Plant leaves.

[Figure]

[Figure]

[Figure]

**Figure 2. Plant leaves: (a) Tribulus leaves, (b) Hippophae leaves, (c) Elm leaves.**

**Point 17:** Figure 6: the mean (black square) is hidden behind the dots. Please make it bigger. What are the dots?

**Response:** We are grateful to you for pointing out details in the manuscript. I totally agree with you, and we will revise Figure 6 in the manuscript by enlarging the mean (black square) to make it easier for the reader to read. These black dots are the amount of daily dew for the leaves of the three plants in May-October, 2022.

**Point 18:** Table 2: Explain what does it mean the columns name. For example: "Dm/mm".

**Response:** We are very grateful to you for pointing out the details in the manuscript. We will explain the meaning of the column names in the notes to Table 2.

**Point 19:** Line 216: I think they are not "factors" they are "atmospheric conditions".

**Response:** We agree with your suggestion. Since Section 3.5 focuses on the analysis of individual factors, we use "factors" instead of "conditions".

**Point 20:** Line 218-222: The paragraph seems to be contradictory. Dew frequency decreases with high wind speed but this wind speed increases the dew amount? How can be possible that dew frequency increases and dew amount decreases?

**Response:** We are very grateful to your valuable comments. Too much or too little wind can limit dew formation, while moderate wind speeds favor dew formation (Vuollekoski et al., 2015; Zhang et al., 2015; Beysens, 2016; Zhuang and Zhao, 2017). The point of this paragraph (Line 218-222) is that the average amount of dew increases with increasing wind speed up to a wind speed of 0.5 m/s, decreases at wind speeds greater than 0.5 m/s; At wind speeds less than 1.0 m/s, the frequency of dew occurrence does not change much. At wind speeds greater than 1 m/s, dew hardly occurs.

**Point 21:** Figure 8: I do not see any correlation between wind speed and dew amount. It need more explanation of why correlated wind speed when it apparently does not play any direct role on dew. $R^2$ and curve slope are not included. Authors should find another statistical parameters to relate wind speed and dew.

**Response:** We gratefully appreciate your comment. It has been well documented that suitable wind speed favours dew formation (Vuollekoski et al., 2015; Zhang et al., 2015; Beysens, 2016; Zhuang and Zhao, 2017), which is consistent with our study. In Fig. 8(d) of the manuscript, we used a Gaussian function to fit the wind speed and the amount of dew, and the fitting results are as follows (Tab.1). We will add the relevant statistical parameters in the manuscript

**Table 1. Results of Gaussian fitting for wind speed and mean dew amount.**

| Typical Plant | Tribulus | Hippophae | Elm |
|---|---|---|---|
| Fit Function | Gaussian function | | |
| Functional Equation | $y = y_0 + \dfrac{A}{w \times \sqrt{\dfrac{\pi}{2}}} \times e^{\frac{-2(x-x_c)^2}{\omega^2}}$ | | |
| $y_0$ | $0.05533 \pm 0.00742$ | $0.01452 \pm 0.00725$ | $0.02512 \pm 0.00189$ |
| $x_c$ | $0.45001 \pm 0.04764$ | $0.41815 \pm 0.03811$ | $0.39924 \pm 0.02486$ |
| $w$ | $1.11006 \pm 0.15028$ | $0.8992 \pm 0.19355$ | $0.37519 \pm 0.06013$ |
| $A$ | $0.10297 \pm 0.02084$ | $0.06294 \pm 0.01874$ | $0.00972 \pm 0.00208$ |
| $R^2$ | $0.94441$ | $0.97032$ | $0.89249$ |

**Point 22:** 3.5.2 Wind direction: I think that analyzing wind direction does not make any sense since wind speed is extremely low. When wind speed is lower than 2 m/s, wind direction is random. Instead of wind speed and direction, authors must include measurements of atmospheric turbulence, which are more related to dew formation (stable boundary layer)

**Response:** We are very grateful to your valuable comments. In the study of the environmental factors of dew characteristics in Yokoyama et al (2021), the wind direction was also analysed. Within 2 m/s, the wind direction has been changing, but it also reflects the main wind direction of the frequency of dew occurrence, which is also of some reference significance. Due to the limitation of the test conditions, atmospheric turbulence was not measured in this test, and we are willing to explore and further study to adopt your suggestion in the subsequent research.

**Point 23:** Line 245: Here is main main issue of the paper. Dew is a surface processes resulted from the surface radiative loss. This radiative loss cool down the surface temperature close or below the dew point, condensing atmospheric water vapor over the surface. If air temperature is close the dew point, it

does not mean that dew is present. It means that air is condensing, resulting in fog. Then, it is not possible to relate dew formation over different leaves surfaces using air temperature. The authors should use leave surface temperature instead of air temperature.

**Response:** We sincerely appreciate your valuable comments and we agree with your views. Due to the limitation of experimental conditions, the temperature of plant leaf surface was not measured in this experiment. A decrease in air temperature will reduce the temperature of the condensation surface, and the same reference exists for using the difference between air temperature and dew point to reflect the conditions of dew occurrence, and the method is reflected in some literatures (Clus et al., 2008; Maestre-Valero et al., 2011; Beysens, 2016; Xie et al., 2021).

**Point 24:** Line 295: water vapor from where?

**Response:** We gratefully appreciate your comment. The experimental station is located in the loess hills and gullies area, where the mountain wind is dominant. The main wind direction is 290° to 330°, and water vapor mainly comes from the main wind direction.

**Point 25:** Line 300 to 315: I think authors should include data of the leave roughness and inclination angle in the results section to after discuss here they hypothesis. Note that leave roughness also can be favourable to collect fog.

**Response:** We fully agree with your suggestion that it helps to enrich the content and increase the quality of the manuscript. We will add the data on leaf inclination and leaf roughness to the manuscript.

**Point 26:** Line 400: Consider to change the title, which is very similar to the one in line 400.

**Response:** We are very grateful to your valuable comment. We take this issue you raised very seriously and we will make changes to the title.

**References**

Beysens, D.: Estimating dew yield worldwide from a few meteo data, Atmos. Res., 167, 146-155, https://doi.org/10.1016/j.atmosres.2015.07.018, 2016.

Clus, O., Ortega, P., Muselli, M., Milimouk, I., and Beysens, D.: Study of dew water collection in humid tropical islands, Journal of Hydrology, 361, 159-171, https://doi.org/10.1016/j.jhydrol.2008.07.038, 2008.

Maestre-Valero, J. F., Martínez-Alvarez, V., Baille, A., Martín-Górriz, B., and Gallego-Elvira, B.: Comparative analysis of two polyethylene foil materials for dew harvesting in a semi-arid climate, Journal of Hydrology, 410, 84-91, https://doi.org/10.1016/j.jhydrol.2011.09.012, 2011.

Jia, Z. F., Wu, B., Wei, W., Chang, Y., Lei, R., Hu, W., and Jiang, J.: Effect of Plastic Membrane and Geotextile Cloth Mulching on Soil Moisture and Spring Maize Growth in the Loess–Hilly Region of Yan'an, China, Agronomy, 13, https://doi.org/10.3390/agronomy13102513, 2023.

Wang, W. K., Zhang, Z. Y., Yin, L. H., Duan, L., and Huang, J.: Topical Collection: Groundwater recharge and discharge in arid and semi-arid areas of China, Hydrogeol. J., 29, 521-524, https://doi.org/10.1007/s10040-021-02308-0, 2021.

Xie, J. J., Su, D. R., Lyu, S. H., Bu, H., and Wo, Q.: Dew formation characteristics of meadow plants canopy at different heights in Hulunbuir grassland, China, Hydrol. Res., 52, 558-571, https://doi.org/10.2166/nh.2021.168, 2021.

Yokoyama, G., Yasutake, D., Wang, W. Z., Wu, Y. R., Feng, J. J., Dong, L. L., Kimura, K., Marui, A., Hirota, T., Kitano, M., and Mori, M.: Limiting factor of dew formation changes seasonally in a semiarid crop field of northwest China, Agric. For. Meteorol., 311, https://doi.org/10.1016/j.agrformet.2021.108705, 2021.

Zhang, Q., Wang, S., Yang, F. L., Yue, P., Yao, T., and Wang, W. Y.: Characteristics of dew formation and distribution, and its contribution to the surface water budget in a Semi-arid Region in China, Bound.-Layer Meteor., 154, 317-331, https://doi.org/10.1007/s10546-014-9971-x, 2015.

Zhang, Q., Wang, S., Yue, P., and Wang, S.: Variation characteristics of non-rainfall water and its contribution to crop water requirements in China's summer monsoon transition zone, Journal of Hydrology, 578, https://doi.org/10.1016/j.jhydrol.2019.124039, 2019.

Zhuang, Y. L. and Zhao, W. Z.: Dew formation and its variation in Haloxylon ammodendron plantations at the edge of a desert oasis, northwestern China, Agric. For. Meteorol., 247, 541-550, https://doi.org/10.1016/j.agrformet.2017.08.032, 2017.